# Jasmonic and Salicylic Acids Enhance Biomass, Total Phenolic Content, and Antioxidant Activity of Adventitious Roots of *Acmella radicans* (Jacq.) R.K. Jansen Cultured in Shake Flasks

**DOI:** 10.3390/biom13050746

**Published:** 2023-04-26

**Authors:** Antonio Bernabé-Antonio, Clarisa Castro-Rubio, Raúl Rodríguez-Anda, José Antonio Silva-Guzmán, Ricardo Manríquez-González, Israel Hurtado-Díaz, Mariana Sánchez-Ramos, Gabriela Hinojosa-Ventura, Antonio Romero-Estrada

**Affiliations:** 1Department of Wood, Pulp and Paper, University Center of Exact Sciences and Engineering, University of Guadalajara, Km 15.5 Guadalajara-Nogales, Col. Las Agujas, Zapopan 45200, Jalisco, Mexico; clarisacastrorubio@yahoo.com.mx (C.C.-R.); raul.randa@academicos.udg.mx (R.R.-A.); jantonio.silva@academicos.udg.mx (J.A.S.-G.); ricardo.manriquez@academicos.udg.mx (R.M.-G.); israel.hurtado@academicos.udg.mx (I.H.-D.); 2Department of Biotechnology, Autonomous Metropolitan University-Iztapalapa Campus, Av. Ferrocarril de San Rafael Atlixco 186, Col. Leyes de Reforma 1^a^. Sección, Alcaldía Iztapalapa, Mexico City 09310, Mexico; marianasan_06@xanum.uam.mx; 3Department of Chemical Engineering, University Center of Exact Sciences and Engineering, University of Guadalajara, Blvd. Marcelino García Barragán 1451, Col. Olímpica, Guadalajara 44430, Jalisco, Mexico; gabriela.hinojosa@academicos.udg.mx

**Keywords:** plant growth regulators, in vitro cultures, adventitious roots, elicitors, wild plant

## Abstract

*Acmella radicans* (Asteraceae) is a plant native to America. Despite it having medicinal attributes, studies on its phytochemical properties are scarce, and biotechnological studies do not exist for this species. In this study, we established an adventitious root culture from *A. radicans* internodal segments in shake flasks with indole-3-butyric acid (IBA), and then elicited it with jasmonic acid (JA) and salicylic acid (SA). The total phenolic content and antioxidant activity were evaluated, and a comparison was made using in vitro plantlets and wild plants. Internodal segments with 0.1 mg/L IBA showed 100% root induction and exhibited better growth after transfer to shake flasks with MS liquid culture medium. JA had a significant effect on biomass increase compared to unelicited roots, mainly with 50 µM JA (28%), while SA did not show significant results. Root elicited with 100 µM (SA and JA) showed a 0.34- and 3.9-fold increase, respectively, in total phenolic content (TPC) compared to the control. The antioxidant activity was also significant, and a lower half-maximal inhibitory concentration (IC_50_) was observed as the AJ concentration increased. Roots elicited with AJ (100 µM) exhibited high antioxidant activity with DPPH (IC_50_ = 9.4 µg/mL) and ABTS (IC_50_ = 3.3 µg/mL) assays; these values were close to those for vitamin C (IC_50_ = 2.0 µg/mL). The TPC and antioxidant activity of in vitro plants and root cultured in shake flasks showed the lowest values in most cases; even the root cultures without elicitation were better than those of a wild plant. In this study, we demonstrated that *A. radicans* root culture is capable of producing secondary metabolites, while its production and antioxidant activity can be enhanced using jasmonic acid.

## 1. Introduction

Medicinal plants have been used by people since ancient times due to their wide curative attributes [1]. The Asteraceae family is among the largest in the Mexican flora, and its ethnobotanical and ethnopharmacological reports have revealed therapeutic use due to its bioactive secondary metabolite diversity [2,3]. Among this plant family, some genera such as *Acmella* (formerly listed as *Spilanthes*) [4,5,6] stand out for producing several secondary metabolites, e.g., aliphatic alkamides in roots and flowers, which have analgesic effects on the teeth and throat [7,8,9], while polyphenols have anti-inflammatory, antiseptic, and antioxidant properties [10,11].

In particular, *A. radicans*, commonly named “toothache plant”, is an important species used in traditional Mexican medicine and is distributed across most Mexican states [12]. According to the Mexican Official Standard (NOM-059-SEMARNAT-2010), *A. radicans* is not listed as an endangered species. However, it is necessary to conduct viable alternative studies for its sustainable medicinal use while conserving this genetic resource [13]. To our knowledge, there are only two studies on this species despite its medicinal value. Gas chromatography analysis of the aerial part and roots of *A. radicans* showed that its chemical composition mainly comprises 2-tridecanone, 1-pentadecene, *trans*-*β*-caryophyllene, elemol, guaiol, 2-pentadecanone, 2-pentadecanol, 1-tridecene, and palmitic acid [14]. In another study, the chemical compositions of *A. radicans* ethanolic extracts were compared, and gas chromatography–mass spectrometry (GC–MS) analysis showed a greater amount of alkamides in roots than in the aerial part of intact plants; in contrast, plantlets grown under in vitro conditions without plant growth regulators (PGRs), presented a higher amount of alkamides in the aerial part than in the roots [7]. These studies indicated that the in vitro *A. radicans* adventitious root cultures were capable of producing alkamides.

On the other hand, adventitious root cultures have been reported as being more promising than hairy roots for biomass production due to their rapid growth and stable metabolite productivity [15,16]. Moreover, adventitious roots do not need genetic modification, and compound extraction is more efficient, simple, and safe [17]. In addition, plant hormones such as jasmonic acid (JA) and salicylic acid (SA) play an important role in plants’ secondary metabolism and defense, serve as elicitors in exogenous applications, and stimulate the accumulation of secondary metabolites, i.e., polyphenols [18,19]. Furthermore, adventitious roots elicited with JA or SA can enhance polyphenolic compound production and antioxidant activity [20,21,22,23,24].

This study aimed to establish an adventitious *A. radicans* root culture, add JA and SA elicitors, and evaluate its effect on total phenolic content and antioxidant activity. We also made a comparison between wild plant and in vitro culture extracts.

## 2. Materials and Methods

### 2.1. Plant Material

Whole, mature *A. radicans* plants (roots and aerial part, including flowers and seeds) were collected on 20 December 2018, in Nextipac, Zapopan, Jalisco, México. Geographical coordinates are 20°43′05.31″ north latitude, 103°32′17.55″ west longitude, at 1665 m above sea level. Taxonomic identification was performed at the Laboratorio Nacional de Identificación y Caracterización Vegetal (LaniVeg-CONACYT) and the Herbario “Luz Maria Villareal de Puga” of the Instituto de Botanica of the Universidad de Guadalajara, México (IBUG herbarium). The specimen was confirmed as *A. radicans* (Jacq.) R.K. Jansen (Asteraceae) and deposited with registration No. 209659.

### 2.2. Obtaining Aseptic Plants

The seeds were disinfected by washing with a 2% (*v*/*v*) Hyclin-Plus Neutral detergent (Hycel^®^) for 5 min and then immersed in 70% ethanol for 30 s. Finally, seeds were disinfected with a NaClO solution (20% *v*/*v*) plus Tween^®^ 20 (4 drops/100 mL) for 20 min under constant stirring. Semi-solid MS [25] basal medium was prepared without PGRs, combined with sucrose (30 g/L), pH 5.8, and gelled with 2 g/L of Phytagel^®^ (Sigma, St. Louis, MO, USA). Culture medium (25 mL) was deposited into Gerber^®^ flasks and autoclaved at 121 °C, 15 psi for 18 min. Five aseptic seeds were placed in each flask and incubated at 25 ± 2 °C for a 16 h photoperiod with white fluorescent light at a light intensity of 60 µmol m^−2^ s^−1^ and for 8 h in darkness. The 30-day-old plants were used as nodal and internodal segment sources for plant proliferation and root induction, respectively.

### 2.3. Adventitious Root Induction

To induce root formation, solid MS medium was supplemented with 30 g/L sucrose and different indole-3-butyric acid (IBA: 0.01, 0.1, 1.0, and 2.0 mg/L) and kinetin (KIN: 0.1 and 1.0 mg/L) concentrations. A control without PGRs was also used. Internodal segments approximately 5 mm in length were cut and placed horizontally in Gerber flasks with 25 mL of culture medium containing IBA and KIN. Incubation conditions were similar, as previously described. Each treatment consisted of 4 flasks, each with four internodal segments, and the experiment was repeated twice. The roots obtained from the best induction treatment were used to establish the root culture in shake flasks.

### 2.4. Establishment of Roots Culture in Shake Flasks

Several 250 mL Erlenmeyer flasks containing 80 mL of MS medium (30 g/L sucrose, 0.1 mg/L IBA, and pH at 5.8) were used. Each flask was inoculated with 1.5 g of fresh 15-day-old roots and incubated on an AGO-6040 orbital shaker (Prendo^MR^) at 115 ± 1 rpm at 25 ± 2 °C, over a 16 h photoperiod with white fluorescent light, a light intensity of 60 µmol m^−2^ s^−1^ and for 8 h in darkness. After culturing for 15 days, the roots were harvested, which exhibited a hard and brittle texture; therefore, we decided to evaluate the culture medium’s macronutrient proportions (25, 50, 75, and 100%). Roots grown with macronutrients reduced to 50% showed better appearance; therefore, growth kinetics were performed under these conditions.

### 2.5. Adventitious Roots Growth Kinetics

Several 250 mL flasks with 80 mL of sterile MS liquid culture medium (using macronutrients reduced to 50%, 30 g/L sucrose, and 0.1 mg/L IBA) were inoculated with 1.5 g of roots and incubated in an orbital shaker under the same conditions described in Section 2.4. Due to slow root growth, samples were taken every third day during the first nine days and, subsequently, every six days until a decrease in the biomass dry weight was observed. Each sampling consisted of harvesting the biomasses contained in 3 flasks, washing them with distilled water, drying them in an oven at 45 ± 2 °C, and recording the dry weight. The experiment was performed in duplicate and dry weight data were used to plot the growth curve.

### 2.6. Elicitation of Roots with Jasmonic and Salicylic Acids

Several 250 mL flasks containing 80 mL of MS liquid culture medium (macronutrients reduced to 50%, 30 g/L sucrose, and 0.1 mg/L AIB) were inoculated with 1.5 g of 15-day-old roots and then incubated under the same conditions as in Section 2.4. At 15 days of growth, root cultures were treated with different jasmonic acid (JA: 1, 10, 50, and 100 µM) or salicylic acid (SA: 10, 50, 100, and 300 µM) concentrations.

Cultures without JA or SA were used as a negative control. JA or SA were previously dissolved in absolute ethanol (99.5%) and different solutions were prepared for each concentration and sterilized with sterile syringe filters (Nalgen™; 0.22 μm and 4 mm diameter nylon membranes). The maximum culture medium ethanol concentration was 0.2% (*v*/*v*). After applying the treatments, cultures were allowed to grow for 72 h and then harvested. Biomass was washed with distilled water, dried in an oven at 45 ± 2 °C and stored until use. Each treatment consisted of five flasks and each experiment was repeated twice. Biomass from each treatment was used to obtain ethanolic extracts.

### 2.7. Obtaining Extracts from In Vitro Cultures and Wild Plants

Ethanolic extraction was performed using different biomass resources, i.e., roots elicited at different concentrations of JA; roots elicited at different concentrations of SA; non-elicited roots; in vitro plantlets without PGRs: roots and aerial parts; and wild plants: roots, aerial parts, and flowers. All samples were macerated using 99.5% ethanol at room temperature (25 ± 2 °C) for 48 h, followed by sonication for 30 min. The extraction process was performed twice for each biomass under the same conditions. The extracts were filtered and concentrated under reduced pressure at 40 °C using an EL–131 Rotavapor (Büchi Labortechnik AG, Flawil, Switzerland) equipped with a Whaterbath 461 to remove excess solvent. All extracts were used to determine the total phenolic content and assess antioxidant activity.

### 2.8. Determination of Total Phenolic Content

Total phenolic content (TPC) was determined using the Folin–Ciocalteu method [26]. One hundred microliters of extract dissolved in ethanol (2.5 mg/mL) was briefly mixed with 200 μL of Folin–Ciocalteu phenol reagent (2 N) (Sigma-Aldrich, Inc., St. Louis, MO, USA) and 2.0 mL distilled water, and then incubated under dark conditions at 25 °C for 3 min in an AGO-6040 orbital shaker (Prendo^MR^) at 115 ± 1 rpm. Next, 1.0 mL Na_2_CO_3_ (20%; *p*/*v*) was added and the mixture was incubated again for 1 h. Samples were read against a blank using a DR 5000™ UV–VIS spectrophotometer at 765 nm. TPC was quantified using a calibration curve with gallic acid (GA) reagent from 6.25 to 500 mg/L. The results are expressed in terminus of gallic acid equivalents as mg GAE/g dry extract. All determinations are reported as the mean ± standard deviation of three replicates (*n* = 3).

### 2.9. Evaluation of Antioxidant Activity

#### 2.9.1. DPPH Assay

The evaluation using a 2,2-Diphenyl-1-Picrylhydrazyl (DPPH) assay was carried out following the methodology reported by Debnath et al. [27]. A 1650 µL extract aliquot (1.95–500 µg/mL) or positive control (ascorbic acid: 0.625–5 µg/mL) dissolved in ethanol was briefly mixed with 1650 µL of DPPH (0.1 mM) dissolved in ethanol. The mixture was vigorously shaken for 1 min in a vortex, and then incubated under dark conditions at 25 °C for 30 min in an AGO-6040 orbital shaker (Prendo^MR^) at 115 ± 1 rpm. Samples were read using a DR 5000™ UV–VIS spectrophotometer at 517 nm. Results are media from 3 replicates (*n* = 3) and are expressed as the mean half-maximal inhibitory concentration (IC_50_).

#### 2.9.2. ABTS Assay

Antioxidant activity analysis using 2,2-azino-bis-3-ethylbenzothiazoline-6-sulfonic (ABTS) acid assays was performed according to Debnath et al. [27]. A stock solution was prepared by mixing an ABTS solution (7 mM) with a potassium persulfate solution (K_2_S_2_O_8_: 2.45 mM) at a 1:1 ratio. The mixture was stirred at 116 ± 1 rpm in the dark at 25 °C for 12–16 h. The ABTS^+^ working solution was obtained with a diluted (1:41) ABTS^+^ stock solution using phosphate-buffered saline (PBS, pH 7.4) as a diluent, adjusting its absorbance to 0.7 ± 0.02 at 734 nm. For each extract or ascorbic acid concentration, respective stock solutions were prepared to obtain the final required concentration. Then, a 330 µL ethanol aliquot containing 1.56 to 100 µg/mL extract or 0.625 to 5 µg/mL ascorbic acid was mixed with 2970 µL ABTS^+^ working solution and shaken vigorously. It was subsequently incubated for 6 min under the conditions previously mentioned. Finally, samples were read at 734 nm. Results are media from 3 replicates (*n* = 3) and are expressed as the mean half-maximal inhibitory concentration (IC_50_).

### 2.10. Statistical Analysis

All experiments were subjected to analysis of variance (ANOVA) and Tukey’s multiple comparison analysis method tests (*p* ≤ 0.05) using the statistical program SAS 9.0. Values are expressed as the mean ± standard deviation (SD). All experiments were repeated 2 to 3 times.

## 3. Results and Discussion

### 3.1. In Vitro Plant Culture

*A. radicans* seeds exhibited 100% asepsis. Seed germination began 10 days after sowing, while the maximum germination percentage (50%) was at 15 days. Plantlets showed normal morphological characteristics and had better development after being transferred to larger jars (Figure 1A,B). For plantlet proliferation from nodal explants, PGR addition was not necessary in this study, which showed good root development at 10 days (Figure 1C). Plantlets in 1 L jars reached their maximum height (≈18 cm) at 30 days (Figure 1D). It is possible that nodal segments rooted easily due to the endogenous auxin content produced in the shoots or nodes, since leaves or shoots have been reported as known auxin production sites [28]. It has also been reported that adventitious root formation is regulated by factors such as light; temperature; nutrient availability; and, mainly, wounds, which generate endogenous hormonal signaling; and therefore, induce the de novo rooting process, especially when the shoot organ separates from the root [29].

It has been reported that *A. radicans* propagation under normal conditions (germination in soil) has a low germination rate [7]. In other related species, such as *Spilanthes acmella*, 30% germination has been reported under in vitro conditions after culturing in MS medium for 15 days [30]. In another study, germination percentages of 14, 32, and 72% were reported for *Spilanthes oleraceae*, *S. calva*, and *S. paniculata*, respectively, although it is mentioned that they did not exhibit good development [31].

### 3.2. Adventitious Roots and Callus Induction from Internodal Segments

Formation and root growth are generally promoted by auxins combined with cytokinins [32]. However, the response varies depending on the genotype and explant type [33]. In fact, it has been reported that a high auxin/cytokinin ratio or auxin alone favors the presence of calluses [34]. Other reports indicate that simultaneous root and callus formation may also occur depending on the explant and the type of plant growth regulator [35,36,37]. In our study, *A. radicans* internodal segments without plant growth regulators (control) had no morphogenetic response (Figure 2A), while AIB and KIN addition stimulated three morphogenetic responses, i.e., friable callus (Figure 2B), adventitious roots (Figure 2C), and simultaneously formed friable callus and root (Figure 2D).

Tukey’s multiple range test (*p* ≤ 0.05) showed statistically significant differences in morphogenetic responses (Table 1). The root induction alone was obtained with 0.01 mg/L IBA, regardless of the KIN concentration, obtaining 66.7–75.0% values; however, the multiple comparison test showed that the tree treatments that only induced roots were statistically similar (*p* ≤ 0.05).

The callus induction alone was observed with 1.0 mg/L IBA and 1.0 mg/L KIN, and with 2.0 mg/L IBA regardless of KIN concentration; however, the maximum percentage was 25%. On the other hand, the friable callus and root’s simultaneous formation occurred with mainly 0.1 or 1.0 mg/L IBA (100.0%); however, using 2.0 mg/L, the percentages were lower (75.0–87.5%) (Table 1). In all cases, calluses were friable but did not exhibit successful growth, i.e., a week after the first subculture they turned brown and subsequently died. *A. radicans*-inducted calluses did not grow suitably, probably because callus induction and development require higher auxin concentrations, as observed in Table 1, although this also depends on the genotype and auxin type. In addition, to a certain extent, IBA is thermolabile and decomposes during autoclaving and under light, so sometimes it is necessary to use higher concentrations, or, in this case, to use more effective and stable synthetic auxins such as naphthaleneacetic acid (NAA) and 2,4-dichlorophenoxyacetic acid (2,4-D) [38].

On the other hand, the number of roots per segment was recorded independent of morphogenetic response (roots alone or simultaneous root and callus formation). The mean comparison test showed statistically significant differences (*p* ≤ 0.05) in the number of roots. Although roots alone were obtained at low IBA concentrations of (0.01 mg/L), these treatments induced a low number of roots (≤2.8), while the number of roots increased higher IBA concentrations (≥0.1 mg/L). The maximum number of roots was obtained with 1.0 mg/L IBA (7.4 roots/segment); however, when these were separated from explant and subcultured, they did not have suitable growth and were slightly brown (Figure 2E). Furthermore, all treatments with more than five roots per segment were subcultured to fresh culture medium; however, only the roots with 0.1 mg/L IBA without kinetin had better appearance, better growth, and no more calluses were formed (Figure 2F).

In this regard, there are no reports on *A. radicans*; however, in other close species, i.e., *S. acmella*, a 70% root induction was reported using 0.5 mg/L IBA, and the roots were 6.0 cm long on average at 20 days of culture [39]. Another study on *S. acmella*, high root induction percentages (95.5%) were obtained with 1.0 mg/L IBA in half-strength MS medium, obtaining 10 3 cm roots per shoot [40]. High root induction percentages (100%) have also been reported for *S. oleraceae* and *S. calva* with 0.02 mg/L IBA [31]. Other studies reported callus induction (80%) for *S. acmella* using internodal explants with 3.0 mg/L 2,4-D [41]. Using *S. acmella* leaf explants, 75% callus induction was obtained using 1.5 mg/L 2,4-D [42], while callus induction was 70.5% when using 1.5 mg/L 2,4-D and 0.5 mg/L of benzyl aminopurine (BAP) [39]. Calluses were produced for *S. paniculata* leaf explants using 0.15 mg/L NAA and 1.5 mg/L BA [43]. This indicates that callus induction depends on the explant and the type of growth regulator used, as well as the concentration and combination of growth regulator type [39].

Regarding simultaneous root and callus formation, this has been reported in leaf, petiole, and *Morus alba* L. stem explants in MS medium with 2,4-D [33]. The authors mentioned that the leaf callus formed roots and these increased with the increase in auxin concentration. Root development was achieved in 7.14% of petiole explants treated with 2.0 mg/L 2,4-D and callus formation was observed from the root later on. Roots and calluses also formed in the stems in the treatments with 1.0 and 2.0 mg/L 2,4-D, but the roots formed from the explant and not from the callus. Another study on *Morus alba* L. reported that stem explants formed calluses after 30 days of cultivation with 2,4-D (2.0 mg/L); in addition, there was an induction of thick roots and root hairs from the calluses and explant [37].

### 3.3. Adventitious Roots Cultures in Shake Flasks

Shake flasks, or Erlenmeyer flasks with liquid culture medium, are commonly used for plant cell culture, generally applied for medium optimization or process evaluation, as they are easy to handle [44,45]. In our study, MS medium modified with 25, 50, 75, and 100% macronutrients had no significant effect (*p* > 0.8694) on *A. radicans* root dry weight compared to results reported for other *Spilanthes* species [7,46]. *A. radicans* root dry weights were 2.5 to 2.7 g/L after 15 days of culture; however, roots with macronutrients reduced to 50% showed more vigor and were longer, finer, softer in texture and green. Subsequently, growth kinetics were performed under these conditions and the culture was monitored until the stationary phase. The lag phase lasted 3 days, and the exponential phase then lasted until day 39, where maximum dry biomass accumulation (5.8 g/L) was observed. Finally, the stationary phase was observed, which remained until day 51 (Figure 3A). Until day 15, the roots were more abundant, thin, soft, and greenish (Figure 3B); after 18 days, the roots turned yellowish-green and were thick (Figure 3C). Although growth was exponential after day 18, the roots appeared brownish and thickened.

In general, few studies have been conducted on adventitious root cultures for the Asteraceae family, e.g., *Vernonia amigdalina* roots were grown in MS liquid medium with 2.0 mg/L IBA, whose growth lasted up to six weeks and a maximum of 27 g/L fresh biomass at week five was observed [47]. The highest accumulation of dry root biomass was 10.8 g/L at four weeks for *Withania somnifera* root cultures with 0.5 mg/L IBA and 50% macronutrients [48].

In a study on *Andrographis paniculata*, roots grown with 1.0 mg/L NAA showed the highest dry weight (11.3 ± 1.5 g/L) at week five [49]. In contrast, *Gynostemma pentaphyllum* roots supplemented with 0.5 mg/L IBA and 0.5 mg/L NAA showed the highest biomass (5.7 g/L) at 28 days [50]. In *Prunella vulgaris*, the roots with 1 mg/L NAA exhibited maximum biomass production (2.1 g/L) at 21 days of culture and decreased over the following 49 days [51]. These studies highlight the importance of adventitious roots cultures, which were shown to produce secondary metabolites of medicinal interest, thus contributing to genetic resource conservation. Moreover, studies show that the growth time primarily depends on the species [48], since some species lasted up to six weeks; however, for *A. radicans* roots, this was 51 days, although the roots with this age were noted as being brown, which made them unsuitable for study. Therefore, day 15 was considered the best time to perform the elicitation experiments in *A. radicans* root cultures.

### 3.4. Elicitor Effects on Adventitious Root Cultures in Shake Flasks

#### 3.4.1. SA and JA Effect on Root Dry Weight

Elicitation is a strategy that helps overcome several difficulties associated with the large-scale production of most commercially important bioactive secondary metabolites from wild and cultivated plants or in vitro plant cultures [52]. Salicylic acid (SA) and jasmonic acid (JA) are signaling molecules considered the most used abiotic elicitors [53,54,55]. They induce or enhance the biosynthesis of specific secondary metabolites by inducing defense or stress-induced responses when applied to a living cell culture in small amounts [56].

In our study, statistical analyses showed that the addition of SA had no significant effect (*p* ≤ 0.05) on dry root weight; however, weight decreased when applying 300 µM SA. (Figure 4A); in contrast, JA positively affected dry weight up to 50 µM (3.9 g/L), but this decreased at 100 µM, the highest concentration used (Figure 4B).

In a study on a *Psoralea corylifolia* root crop, there was a decrease in dry weight when 25, 50, and 100 μM JA were used, and there was darkening in the roots from the first week of culture. AS also had a negative effect on root biomass production at 25 µM [57]. In *Echinacea purpurea* (Astereaceae), JA reduced fresh biomass production to 1.0 µM, but increased the amount of alkamides [55]. In contrast, SA at 100 µM did not affect root production in species such as *Cichorium intybus* [58]. A similar behavior occurred in *Artemisia annua* roots, where JA treatments at 50, 100, and 200 µM did not show any significant effect [59]. The results suggest that, compared to other species, adventitious *A. radicans* roots are less sensitive to the JA and SA concentrations used.

On the other hand, changes in root coloration were also observed. SA had no significant effect on the roots, the dry weight and the greenish coloration did not change, but a slightly yellowish coloration was noted at 300 µM SA. Although JA caused increases in root weight, root cultures became browner as the JA increased to 50 and 100 µM. Studies have shown that JA causes damage to chloroplasts, thus stimulating chlorophyll degradation by decreasing photosynthetic activity; in addition, JA also promotes organ growth and is involved defense responses to herbivore attack through secondary metabolite production and [60,61].

#### 3.4.2. SA and JA Effects on Total Phenolic Content and Antioxidant Activity

The statistical analysis (*p* ≤ 0.05) showed significance with respect to the control (non-elicited roots) for total phenolic content (TPC) (Figure 5) and antioxidant activity (Figure 6) due to SA or JA elicitation. Roots elicited with SA had TPC values ranging from 43.9 to 52.4 mg GAE/g extract (Figure 5A), which had values close to the control. These results indicate that the evaluated SA concentrations did not have an important effect on *A. radicans’* TPC. In contrast, JA-elicited roots (>10 μM) showed higher amounts of TPC compared to SA-elicited roots. In addition, an increase in TPC was observed as JA concentration increased, showing maximum TPC production (190.2 ± 0.2 mg GAE/g extract) when 100 μM JA was added (Figure 5B).

The results indicate that JA is a better elicitor than SA for increasing phenolic compound production and antioxidant activity in *A. radicans* roots. Furthermore, biomass production was not significantly affected. In a study on *Stevia rebaudiana*, adventitious roots showed higher TPC accumulation (3.9 ± 0.4 mg GAE/g sample) due to the addition of methyl jasmonate [62]. In another study, it was found that *Codonopsis lanceolata* cord roots had the highest TPC (74.5 mg GAE/g sample) when treated with 20 µM methyl jasmonate [63]. A similar study carried out on cocultured adventitious *E. purpurea* and *Echinacea pallida* roots demonstrated that the addition of methyl jasmonate (25 μM) to root culture increases maximum TPC production (728.2 mg GAE/L) [64]. It has also been found that phytohormones can influence phenolic content [65]. In a buckwheat crop, it was found that treatment with 150 μM JA exposed for 48 or 72 h improved phenolic compound levels in buckwheat shoots compared to those observed in the control and shoots treated with 50 and 100 μM JA, while SA did not affect phenolic compound production [66].

Regarding antioxidant activity, statistical analysis showed significant effects with respect to the control when elicited with SA or JA on root cultures (Figure 6A–D). Root cultures with SA showed IC_50_ values ranging from 44.1 to 52.5 μg/mL for the DPPH assay, while IC_50_ values ranged from 14.3 to 17.1 μg/mL for the ABTS assay (Figure 6A,C). Although the values were significant, they were close to the control (non-elicited roots) and higher than for ascorbic acid. In contrast, JA-treated roots showed high antioxidant activity compared to the control (Figure 6B,D).

The best antioxidant activity from the DPPH was obtained with the root extract elicited with 100 μM JA (IC_50_ = 9.4 μg/mL), which can be considered highly antioxidant compared to ascorbic acid (IC_50_ = 1.7 μg/mL) (Figure 6B). For the ABTS, root extracts with 10, 50, and 100 µM JA showed strong antioxidant activity with IC_50_ values of 5.4 ± 0.0, 3.9 ± 0.1, and 3.3 ± 0.0 µg/mL, respectively (Figure 6C,D). This is relevant since the IC_50_ extract values are very close to ascorbic acid (IC_50_ = 2.0 ± 0.0 μg/mL), a powerful, widely known antioxidant [67]. To date, no elicitation studies have been reported in the genus *Acmella* or related species. However, in studies on *Oplopanax elatus*, methanolic extracts of roots elicited with methyl jasmonate (MeJa) at 200 µM showed strong antioxidant activity using DPPH (IC_50_ = 6.2 µg/mL) [20]. Aqueous *Codonopsis lanceolata* root extracts elicited with 20 µM MeJa also showed antioxidant activity using DPPH (IC_50_ = 24.2 µg/mL), being better than the control. In another species, such as *Momordica charantia* (Cucurbitaceae) roots, similar results have been reported, showing root cultures supplemented with 100 μM JA or SA significantly improved the phenolic compounds compared to non-elicited root cultures; likewise, biomass increased significantly with SA, while it decreased with JA at 100 μM. On the other hand, roots treated with JA and SA showed strong antioxidant activity using DPPH compared to unelicited roots [68].

### 3.5. Total Phenolic Content and Antioxidant Activity of In Vitro Plantlets and Wild Plants

As a complementary and comparative study, we also determined the TPC and antioxidant activity of in vitro *A. radicans* plantlet extracts (roots and aerial parts) and wild plants (roots, flowers, and aerial parts). Statistical TPC analysis showed significant differences (*p* ≤ 0.05) between all extracts (Table 2). Flowers from wild plants showed the highest TPC (43.2 ± 0.4 mg GAE/g extract). The roots of both plants were statistically similar, but they were different for the aerial part.

Regarding antioxidant activity, wild plant root extracts had IC_50_ = 142.9 ± 3.6 μg/mL (DPPH) and IC_50_ = 26.2 ± 0.1 μg/mL (ABTS). Flower extracts showed IC_50_ = 133.5 ± 2.5 μg/mL (DPPH) and IC_50_ = 27.8 ± 0.5 μg/mL (ABTS). On the other hand, root extracts cultured in vitro stood out for having the best IC_50_ values with IC_50_ = 76.0 ± 2.1 μg/mL (DPPH) and 24.2 ± 0.1 μg/mL (ABTS) (Table 2).

For wild plant and in vitro aerial part extracts, it was not possible to determine the IC50 using the DPPH method because the intrinsic color of the extracts had interference at concentrations above 250 μg/mL; however, at this concentration, inhibition percentages of 47.5% were obtained for wild plants and 30.5% for in vitro plants. In fact, it is reported that TPC and the antioxidant activity are dependent on the raw material and the extraction solvent [69].

Similar studies on *Spilanthes ciliate* have been reported, showing higher TPC content (21.5 mg GAE/g of dry biomass) was found in ethanolic leaf extracts than in other plant organs [70]. In *S. acmella* ethanolic leaf extracts, TPC have been reported in amounts as high as 84.5 mg GAE/g extract; moreover, extracts showed antioxidant activity with IC_50_ = 134.1 µg/mL using the DPPH method [71]. For *Acmella oleraceae*, leaves stood out from other organs for containing higher TPC (7.6 mg GAE/g biomass), as well as higher antioxidant capacity in terms of 5.3 mg Trolox equivalents/g biomass [72]. In another study, higher TPC content was also found in *Acmella alba* and *A. oleraceae* leaves, and *Acmella calirrhiza* flowers. In addition, this behavior occurred for antioxidant activity when using DPPH and ABTS [73].

Antioxidant activity is a functional benefit provided by plant extracts because they contain a wide variety of natural antioxidants. Many of these plant extracts and compounds are emerging as candidates for moderating the effects of the aging process on the skin, as well as food additives. In fact, consumer perception of antioxidants is positive, making them particularly attractive as cosmetic ingredients [74]. As demonstrated, in vitro plant cultures can significantly improve antioxidant compound production, which can be sustainably exploited to produce antioxidants while conserving genetic resources. In addition, this work establishes the basis for future studies on *A. radicans* cord root cultivation, for example, scaling up to the bioreactor to generate industrially important secondary metabolites.

## 4. Conclusions

In this work, an adventitious root culture, plants cultured in vitro, and wild plants were evaluated. Studies on non-genetically modified adventitious root cultures have shown great potential for producing secondary metabolites. The *A. radicans* roots were easily induced using low indolebutyric acid concentrations, and roots grown in liquid medium elicited with jasmonic acid showed great potential for producing phenolic compounds with strong antioxidant activity compared to wild plants or unelicited cultures. This indicates that *A. radicans* roots have great potential. Therefore, in future, studies on roots can be scaled up in a bioreactor to further improve the production of biomass and secondary metabolites of industrial interest. This is the first report on the development of *A. radicans* root cultures demonstrating the potential for producing antioxidant compounds, as the results are comparable to those obtained for ascorbic acid.

## Figures and Tables

**Figure 1 biomolecules-13-00746-f001:**
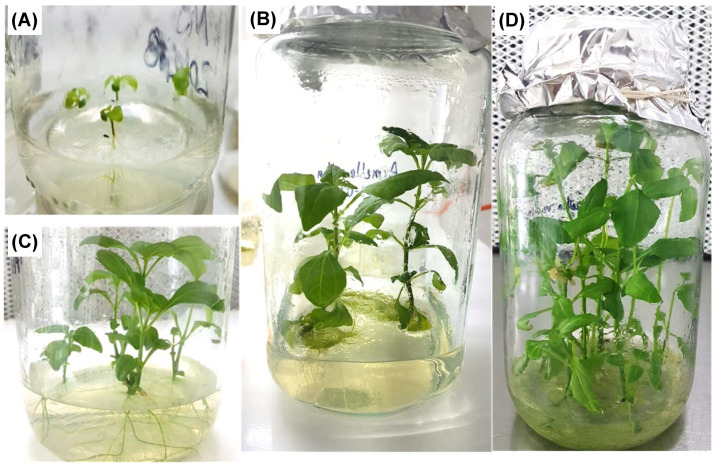
In vitro cultures of *A. radicans* in solid MS culture medium without PGRs. (**A**) Seedlings 10-days-old; (**B**) plants transferred to 1 L flasks; (**C**) nodal explants cultured at 10 days; (**D**) plants from nodal explants cultured at 30 days.

**Figure 2 biomolecules-13-00746-f002:**
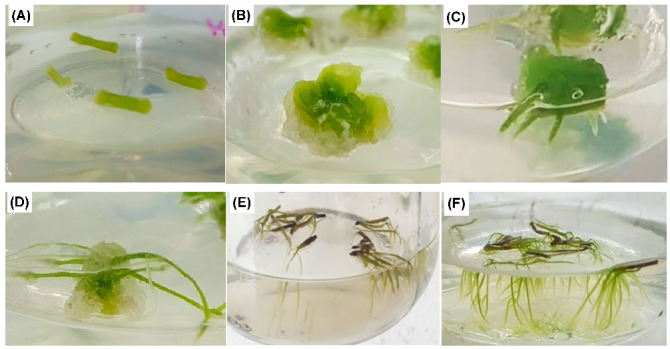
Morphogenetic response from *A. radicans* internodal segments in solid MS medium cultured at 15 days. (**A**) Internodes without plant growth regulators; (**B**) callus induced with 2.0 mg/L IBA and 1.0 mg/L KIN; (**C**) root induced with 1 mg/L IBA; (**D**) simultaneous induction of roots and calluses with 0.1 mg/L IBA; (**E**) isolated roots from internodal segment with 1 mg/L IBA; (**F**) isolated roots from internodal segment with 0.1 mg/L IBA.

**Figure 3 biomolecules-13-00746-f003:**
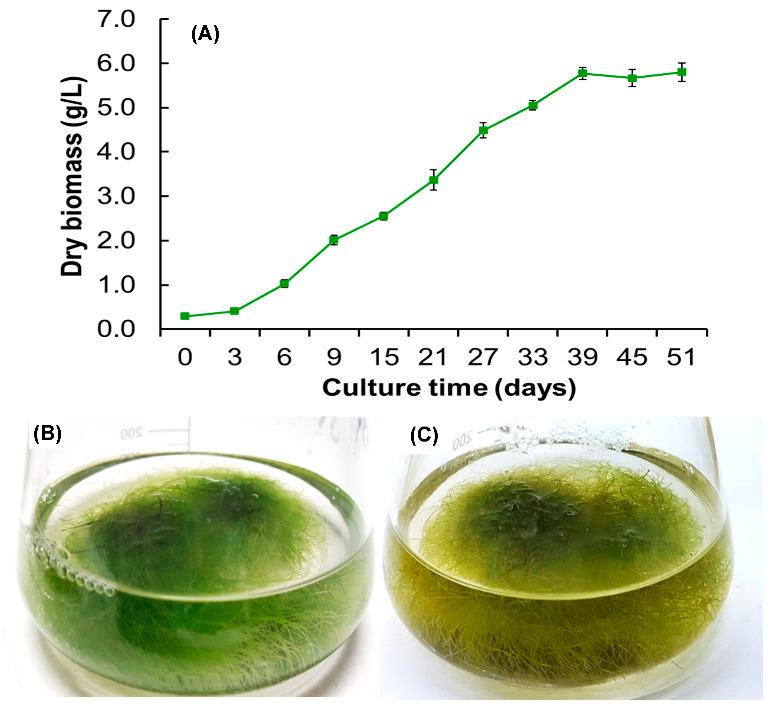
Adventitious *A. radicans* root cultures grown in shake flasks with MS liquid culture medium, 0.1 mg/L IBA, and 50% macronutrients. (**A**) Root culture growth kinetics; (**B**) root cultures at 15 days; (**C**) root cultures at 18 days. Values represent the means ± standard deviation of three replicates.

**Figure 4 biomolecules-13-00746-f004:**
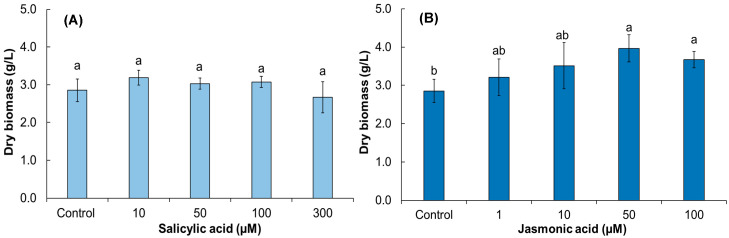
Dry weight of adventitious *A. radicans* root cultures at 18 days. (**A**) Salicylic acid-elicited roots; (**B**) jasmonic acid-elicited roots. Values represent the means ± standard deviation from two independent experiments. Bars followed by the same letters are not significantly different according to Tukey’s multiple range test (*p* ≤ 0.05). Control = non-elicited roots.

**Figure 5 biomolecules-13-00746-f005:**
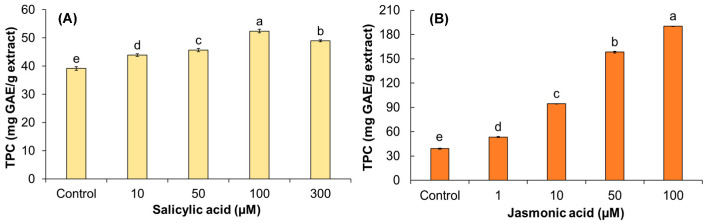
Effect of salicylic (SA) and jasmonic acids (JA) on total phenolic content (TPC) in adventitious *A. radicans* root cultures. (**A**) SA-elicited roots; (**B**) JA-elicited roots. Values represent the means ± standard deviation of three measurements (*n* = 3). Bars followed by the same letters are not significantly different according to Tukey’s multiple range test (*p* ≤ 0.05). Control = non-elicited roots.

**Figure 6 biomolecules-13-00746-f006:**
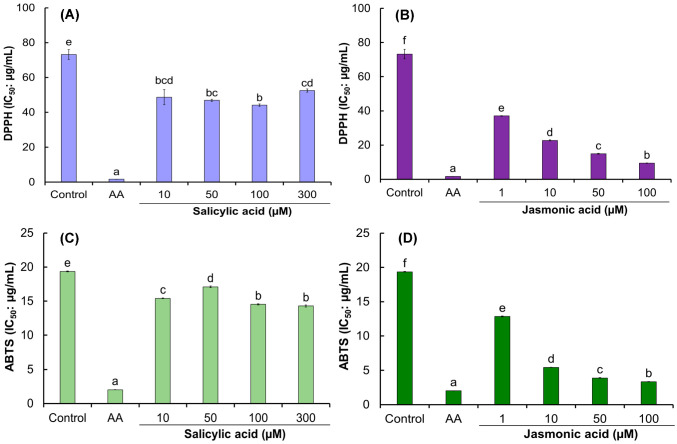
Effect of salicylic (SA) and jasmonic acids (JA) on adventitious *A. radicans* root culture antioxidant activity. (**A**) DPPH assay for SA-elicited roots; (**B**) DPPH assay for JA-elicited roots; (**C**) ABTS assay for SA-elicited roots; (**D**) ABTS assay for JA-elicited roots. Values represent the means ± standard deviation of three measurements (*n* = 3). Values followed by the same letter are not statistically different (Tukey, *p* ≤ 0.05). AA—ascorbic acid. Control = non-elicited roots.

**Table 1 biomolecules-13-00746-t001:** IBA’s and KIN’s effect on *A. radicans* internodal segment morphogenetic responses in solid MS medium after culturing for 30 days.

PGRs (mg/L)		Roots Induction (%)	Callus Induction (%)	Simultaneous Root and Callus Induction (%)	No. of Roots per Segment
IBA	KIN				
0.0	0.0	0.0 ± 0.0 ^b^	0.0 ± 0.0 ^c^	0.0 ± 0.0 ^d^	0.0 ± 0.0 ^f^
0.01	0.0	75.0 ± 20.4 ^a^	0.0 ± 0.0 ^c^	0.0 ± 0.0 ^d^	2.8 ± 0.1 ^cde^
0.01	0.1	66.7 ± 11.8 ^a^	0.0 ± 0.0 ^c^	0.0 ± 0.0 ^d^	1.9 ± 0.2 ^def^
0.01	1.0	66.7 ± 23.6 ^a^	0.0 ± 0.0 ^c^	0.0 ± 0.0 ^d^	1.3 ± 0.5 ^ef^
0.1	0.0	0.0 ± 0.0 ^b^	0.0 ± 0.0 ^c^	100.0 ± 0.0 ^a^	5.0 ± 1.2 ^abc^
0.1	0.1	0.0 ± 0.0 ^b^	0.0 ± 0.0 ^c^	100.0 ± 0.0 ^a^	5.5 ± 0.4 ^ab^
0.1	1.0	0.0 ± 0.0 ^b^	0.0 ± 0.0 ^c^	100.0 ± 0.0 ^a^	3.8 ± 0.9 ^bcd^
1.0	0.0	0.0 ± 0.0 ^b^	0.0 ± 0.0 ^c^	100.0 ± 0.0 ^a^	7.4 ± 1.7 ^a^
1.0	0.1	0.0 ± 0.0 ^b^	0.0 ± 0.0 ^c^	100.0 ± 0.0 ^a^	5.7 ± 1.7 ^ab^
1.0	1.0	0.0 ± 0.0 ^b^	12.5 ± 4.9 ^b^	87.5 ± 4.9 ^b^	3.3 ± 1.5 ^bcde^
2.0	0.0	0.0 ± 0.0 ^b^	12.5 ± 4.9 ^b^	87.5 ± 4.9 ^b^	3.6 ± 0.8 ^bcde^
2.0	0.1	0.0 ± 0.0 ^b^	19.1 ± 4.2 ^ab^	80.9 ± 4.2 ^bc^	4.1 ± 1.2 ^bcd^
2.0	1.0	0.0 ± 0.0 ^b^	25.0 ± 8.2 ^a^	75.0 ± 8.2 ^c^	4.0 ± 0.6 ^bcd^

PGRs—plant growth regulators; IBA—indole-3-butyric acid; KIN—kinetin. Data represent the means ± standard deviation from two independent experiments. Values with the same letters in superscript in column are not significantly different according to Tukey’s multiple range test (*p* ≤ 0.05).

**Table 2 biomolecules-13-00746-t002:** In vitro and wild *A. radicans* ethanolic plant extracts’ total phenolic content and antioxidant activity.

Extract Source	Total Phenolics (mg GAE/g Extract)	IC_50_ (μg/mL)	
		DPPH	ABTS
In Vitro plants *			
Roots	31.2 ± 0.1 ^c^	76.0 ± 2.1 ^b^	24.2 ± 0.1 ^b^
Aerial part	24.0 ± 0.2 ^d^	>250 ^†^	52.5 ± 0.7 ^f^
Wild plants			
Roots	31.8 ± 0.8 ^c^	142.9 ± 3.6 ^d^	26.2 ± 0.1 ^c^
Aerial part	32.8 ± 0.3 ^b^	>250 ^†^	48.4 ± 0.2 ^e^
Flowers	43.2 ± 0.4 ^a^	133.5 ± 2.5 ^c^	27.8 ± 0.5 ^d^
Ascorbic acid	n.a.	1.7 ± 0.1 ^a^	2.0 ± 0.0 ^a^

* Plant growth regulators—free plantlets. ^†^ IC50 could not be determined due to extract coloration interference. n.a.—not applicable. Values represent the means ± standard deviation of three measurements (*n* = 3). Values followed by the same letter in columns are not statistically different (Tukey, *p* ≤ 0.05).

## Data Availability

Data are contained within this article.

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
