# Peer review of "Jasmonic and Salicylic Acids Enhance Biomass, Total Phenolic Content, and Antioxidant Activity of Adventitious Roots of Acmella radicans (Jacq.) R.K. Jansen Cultured in Shake Flasks"

_biomolecules, 2023, doi:10.3390/biom13050746_

Round 1
Reviewer 1 Report
The manuscript entitled “Jasmonic and Salicylic Acids Enhance Biomass, Total Phenolics Content, and Antioxidant Activity of Adventitious Root Suspension Cultures of Acmella radicans (Jacq.) R.K. Jansen”. Authors have established an adventitious root suspension culture from A. radicans internodal segments using indole-3-butyric acid (IBA), and then elicited it with jasmonic acid (JA) and salicylic acid (SA). In this study, The total phenolics content and antioxidant activity were evaluated and compared with in vitro plantlets and wild plants. The results showed that root cultures of A. radicans are capable of producing secondary metabolites and that their production and antioxidant activity can be enhanced with jasmonic acid. The study demonstrates the potential of using root cultures of A. radicans for the production of secondary metabolites with medicinal attributes. However, the manuscript needs some modification. Therefor I recommend this article for further consideration after incorporating changes given in below.
Authors are asked to check the formatting, and use of symbols, etc.,
Acmella radicans name should be full in first mention rest all will be abbreviated as A. radicans check it and revise the same throughout the manuscript.
If possible, provide different clarity images for figure 1B and D.
In figure 2 B: Callus is not clear. Callus is seeming like nodular callus. Whether authors obtain any friable callus?
Figure 2 B, C, D need to enhance the quality.
Line 207: Spilanthes oleraceae, Spilanthes calva and Spilanthes paniculata should be Spilanthes oleraceae, S. calva and S. paniculata
Line 212: Check the lines. Because Auxin and cytokinin intermediate ratio only induce the callus formation. You have mentioned high auxin/cytokinin check it.
In figure 3: Authors should provide the control for root suspension culture.
The manuscript needs thorough proof reading and formatting.
Results and discussion looks shallow. It should be discussed more.
Line 428: In vitro should be italics.
If possible, authors are advised to perform the GC-MS profiling. That is important for your study.
Conclusion section needs to be improvised and future perspectives and hypothesize the current study will also need to be improvised. It will be useful to the readers community to design and understand the importance of studied issue.
Author Response
Response to Reviewer 1 Comments
The manuscript entitled “Jasmonic and Salicylic Acids Enhance Biomass, Total Phenolics Content, and Antioxidant Activity of Adventitious Root Suspension Cultures of Acmella radicans (Jacq.) R.K. Jansen”. Authors have established an adventitious root suspension culture from A. radicans internodal segments using indole-3-butyric acid (IBA), and then elicited it with jasmonic acid (JA) and salicylic acid (SA). In this study, The total phenolics content and antioxidant activity were evaluated and compared with in vitro plantlets and wild plants. The results showed that root cultures of A. radicans are capable of producing secondary metabolites and that their production and antioxidant activity can be enhanced with jasmonic acid. The study demonstrates the potential of using root cultures of A. radicans for the production of secondary metabolites with medicinal attributes. However, the manuscript needs some modification. Therefor I recommend this article for further consideration after incorporating changes given in below.
Point 1. Authors are asked to check the formatting, and use of symbols, etc.,
Response 1. This was done. The format and the use of symbology were revised again throughout the manuscript. Moreover, the manuscript was revised by an English-speaking professional. A certificate is attached.
Point 2. Acmella radicans name should be full in first mention rest all will be abbreviated as A. radicans check it and revise the same throughout the manuscript.
Response 2. This was done. The name Acmella radicans was written out in full at first mention, the rest were abbreviated as A. radicans throughout the manuscript.
Point 3. If possible, provide different clarity images for figure 1B and D.
Response 3. This was done. Figures 1B, 1C, and 1D were substituted and have been enlarged for clarity.
Point 4. In figure 2 B: Callus is not clear. Callus is seeming like nodular callus. Whether authors obtain any friable callus?
Response 4.
Yes, in all cases the callus obtained was friable callus; but since our aim was not to obtain friable callus but adventitious roots, we did not give relevance to callus cultures. We only wished to show the different morphogenetic responses obtained, showing relevance to the obtaining of adventitious roots. However, the friable callus were subcultured in the same culture medium, but they did not show growth, turned brown, and then died. This is described in section 2.3, lines 225, 247-249. However, in future studies we will be reporting a study on friable callus and cell suspension cultures.
Point 5. Figure 2 B, C, D need to enhance the quality.
Response 5. This was done. Figures 2B and 2D were replaced and the quality of Figure 2C was enhanced.
Point 6. Line 207: Spilanthes oleraceae, Spilanthes calva and Spilanthes paniculata should be Spilanthes oleraceae, S. calva and S. paniculata
Response 6. This was done. Line 215.
Point 7: Line 212: Check the lines. Because Auxin and cytokinin intermediate ratio only induce the callus formation. You have mentioned high auxin/cytokinin check it.
Response 7. It is widely reported that in in vitro culture, auxins induce callus or roots formation when there is a higher level of auxins than cytokinins, or auxins alone. The explant response to high, intermediate or low auxin concentrations will depend largely on the genotype and auxin type. Lines 249-250. Even of type and explant age. This is widely reported. For example:
https://doi.org/10.1093/treephys/21.7.457; https://www.revista.ccba.uady.mx/ojs/index.php/TSA/article/view/379/350; https://doi.org/10.55493/5003.v12i1.4449; http://dx.doi.org/10.4067/S0717-92002014000100011.
Point 8. In figure 3: Authors should provide the control for root suspension culture.
Response 8. In this case, it is not possible to show a control of the roots because the roots without growth regulators did not grow in liquid medium. The growth kinetics experiment was performed only with the best root induction treatment as discussed in section 3.2. This is explained in lines 105-109 and 264-266.
Point 9. The manuscript needs thorough proof reading and formatting.
Response 9. This was done. The manuscript was completely revised, and the language was corrected by an English-speaking professional. A certificate is attached.
Point 10. Results and discussion looks shallow. It should be discussed more.
Response 10. This was done. More discussion was given where necessary. Lines 200-206, 247-255, 291-293, 319-324, 385-388, 412-417, 452-461.
Point 11. Line 428: In vitro should be italics.
Response 11. The entire manuscript was reviewed and the word “in vitro” was italicized in all cases.
Point 12. If possible, authors are advised to perform the GC-MS profiling. That is important for your study.
Response 12. Thank you for your advice. Unfortunately, we did not perform the GC-MS for this study. Our main objective was to establish the basis for obtaining an adventitious root culture in vitro, elicit JA and SA and evaluate total phenolics and antioxidant activity, and compare with the wild plant. We believe that this study fulfills the basis for future studies. In fact, in another project, we are currently conducting the compounds isolation by column chromatography and their characterization by NMR) from the wild plant. We are also evaluating other JA and SA concentrations in root cultures and we will be reporting the GC-MS analysis for both cases. These results will be reported in future articles. We can attach a record of our project in progress, if the reviewers request it.
The present study is original and is the first to report in vitro culture, elicitation and antioxidant activity and we believe that it is the basis for future studies.
Point 13. Conclusion section needs to be improvised and future perspectives and hypothesize the current study will also need to be improvised. It will be useful to the readers community to design and understand the importance of studied issue.
Response 13. Conclusions were rewritten as suggested. Lines 466-473.

Reviewer 2 Report
Jasmonic and Salicylic Acids Enhance Biomass, Total Phenolics Content, and Antioxidant Activity of Adventitious Root Suspension Cultures of Acmella radicans (Jacq.) R.K. Jansen. By Antonio Bernabé-Antonio, Clarisa Castro-Rubio, Raúl Rodríguez-Anda, José Antonio Silva-Guzmán, Ricardo Manríquez-González, Israel Hurtado-Díaz, Mariana Sánchez-Ramos, Gabriela Hinojosa-Ventura and Antonio Romero-Estrada.
The manuscript discusses the establishment of an adventitious root suspension culture from A. radicans internodal segments using indole-3-butyric acid (IBA), and then elicitation with jasmonic acid (JA) and salicylic acid.
The biotechnological application of indigenous plants and their parts for socio-economic benefits highlights a promising approach to the utilization of our natural resources for the benefit of human society.
Acmella radicans (Jacq.) R.K. Jansen is a plant negligibly studied for its biotechnological applications. This study defines forms the basis and defines interesting prospects for plant tissue culture-based production and the use of elicitors for enhanced metabolite production. The research aims to investigate and establish a tissue culture system for a less studied plant and improve bioactive metabolite production via the elicitation method.
The present work contributes to the use of less-studied plants (Acmella radicans (Jacq.) R.K. Jansen) and their prospects in a socio-economic context and forms a platform for future studies in this plant towards, considering the beneficial prospects of bioactive metabolites in drug discovery. While, the study also promotes the genetic resources of introducing in-vitro cultures of a less studied plant, and provides a method for the establishment of root suspension cultures.
Specific comments-
Figure 2, please provide a clear resolution image of all figures, the quality of the figure must be improved.
The paper needs extensive revision for the English language for clarity and consistency, for e.g. line 31-Roots elicited with AJ (100 µM) exhibited high anti-oxidant activity for DPPH……………
Line 26: JA had a significant slight biomass increase compared to….the sentence should be rewritten as “JA had a significant effect on biomass increase……….
Line 358: Results indicate that JA is the best elicitor than SA, should be rewritten as “Results indicate that JA is better elicitor than SA……………
Regarding scientific names of the plant species, full-name can be written in the first mention, thereafter abbreviations can be used likewise.
Further studies including GC-MS profiling of plant metabolites and characterization would benefit the quality of the performed work.
The section on ‘Result and Discussion’- is very brief and not addressed in detail. The authors need to present the research outcome in detail, discussing how the present study is beneficial and stands out compared to other research on plant-tissue culture-based investigations and bioactive metabolite production. Future directions and outcomes should be discussed.
References: The references cited in the literature are appropriate in a research context, but they can be improved in regard to similar studies in this context.
The paper needs to revise for English language, grammar, and the use of symbols. The paper is of general importance and can be accepted subject to revisions incorporating all the comments.
Author Response
Response to Reviewer 2 Comments
Jasmonic and Salicylic Acids Enhance Biomass, Total Phenolics Content, and Antioxidant Activity of Adventitious Root Suspension Cultures of Acmella radicans (Jacq.) R.K. Jansen. By Antonio Bernabé-Antonio, Clarisa Castro-Rubio, Raúl Rodríguez-Anda, José Antonio Silva-Guzmán, Ricardo Manríquez-González, Israel Hurtado-Díaz, Mariana Sánchez-Ramos, Gabriela Hinojosa-Ventura and Antonio Romero-Estrada.
Point 1. The manuscript discusses the establishment of an adventitious root suspension culture from A. radicans internodal segments using indole-3-butyric acid (IBA), and then elicitation with jasmonic acid (JA) and salicylic acid.
The biotechnological application of indigenous plants and their parts for socio-economic benefits highlights a promising approach to the utilization of our natural resources for the benefit of human society.
Acmella radicans (Jacq.) R.K. Jansen is a plant negligibly studied for its biotechnological applications. This study defines forms the basis and defines interesting prospects for plant tissue culture-based production and the use of elicitors for enhanced metabolite production. The research aims to investigate and establish a tissue culture system for a less studied plant and improve bioactive metabolite production via the elicitation method.
The present work contributes to the use of less-studied plants (Acmella radicans (Jacq.) R.K. Jansen) and their prospects in a socio-economic context and forms a platform for future studies in this plant towards, considering the beneficial prospects of bioactive metabolites in drug discovery. While, the study also promotes the genetic resources of introducing in-vitro cultures of a less studied plant, and provides a method for the establishment of root suspension cultures.
Response 1. Thanks for your comment.
Point 2. Specific comments-
Figure 2, please provide a clear resolution image of all figures, the quality of the figure must be improved.
Response 2. This was done. Most of figures 1 and 2 were revised and replaced.
Point 3. The paper needs extensive revision for the English language for clarity and consistency, for e.g. line 31-Roots elicited with AJ (100 µM) exhibited high anti-oxidant activity for DPPH……………
Response 3. This was done. The manuscript was completely revised, and the language was corrected by an English-speaking professional. A certificate is attached.
Point 4. Line 26: JA had a significant slight biomass increase compared to….the sentence should be rewritten as “JA had a significant effect on biomass increase……….
Response 4. This was done. The manuscript was completely revised, and the language was corrected by an English-speaking professional. A certificate is attached.
Point 5. Line 358: Results indicate that JA is the best elicitor than SA, should be rewritten as “Results indicate that JA is better elicitor than SA……………
Response 5. This was done. The manuscript was completely revised, and the language was corrected by an English-speaking professional. A certificate is attached.
Point 6. Regarding scientific names of the plant species, full-name can be written in the first mention, thereafter abbreviations can be used likewise.
Response 6. The entire manuscript was reviewed, and the scientific names were corrected as suggested.
Point 7. Further studies including GC-MS profiling of plant metabolites and characterization would benefit the quality of the performed work.
Response 7. Thank you for your advice. Unfortunately, we did not perform the GC-MS for this study. Our main objective was to establish the basis for obtaining an adventitious root culture in vitro, elicit JA and SA and evaluate total phenolics and antioxidant activity, and compare with the wild plant. We believe that this study fulfills the basis for future studies. In fact, in another project, we are currently conducting the compounds isolation by column chromatography and their characterization by NMR) from the wild plant. We are also evaluating other JA and SA concentrations in root cultures, and we will be reporting the GC-MS analysis for both cases. These results will be reported in future articles. We can attach a record of our project in progress if the reviewers request it.
The present study is original and is the first to report in vitro culture, elicitation and antioxidant activity and we believe that it is the basis for future studies.
Point 8. The section on ‘Result and Discussion’- is very brief and not addressed in detail. The authors need to present the research outcome in detail, discussing how the present study is beneficial and stands out compared to other research on plant-tissue culture-based investigations and bioactive metabolite production. Future directions and outcomes should be discussed.
Response 8. This was done. More discussion was given where necessary. Lines 200-206, 247-255, 291-293, 319-324, 385-388, 412-417, 452-461.
Point 9. References: The references cited in the literature are appropriate in a research context, but they can be improved in regard to similar studies in this context.
Response 9. This was done. More discussion was given where necessary. Lines 200-206, 247-255, 291-293, 319-324, 385-388, 412-417, 452-461. See References No. 13, 28, 29, 38, 44,45, 66,67 and 74.
Point 10. The paper needs to revise for English language, grammar, and the use of symbols. The paper is of general importance and can be accepted subject to revisions incorporating all the comments.
Response 10. This was done. The manuscript was completely revised, and the English language was corrected by an English-speaking professional. A certificate is attached.

Reviewer 3 Report
I have completed a review of your above manuscript titled “Jasmonic and Salicylic Acids Enhance Biomass, Total Phenolics Content, and Antioxidant Activity of Adventitious Root Suspension Cultures of Acmella radicans (Jacq.) R.K. Jansen”
Thank you for the opportunity to review this paper regarding the procedure for obtaining in vitro cultures (plants, adventitious roots) of Acmella radicans.
Data from the scientific literature indicate that folk applications may be conditioned by the content of secondary metabolites mentioned by the authors of the work.
The legitimacy of undertaking research in in vitro cultures makes sense in the case of relict plants, protected rare plants, etc. Does the species studied by the authors of the manuscript belong to this category of plants? This was not mentioned in the justification for undertaking the research. However, regardless of whether this plant is rare or more common in the world, the developed in vitro cultures are of practical importance because - among other things, it gives procedures for introducing other, endangered or rare plant species from this genus or closely related to in vitro cultures, and it is a form of safeguarding genetic resources.
Dear Editors, Authors. Before the work goes to print, please respond to my comments point by point. Please pay particular attention to errors in terminology regarding the concept of suspension cultures.
Other comments are included in the form of a commentary in the text (pdf).
Yours faithfully,
reviewer

Author Response
Response to Reviewer 3 Comments
I have completed a review of your above manuscript titled “Jasmonic and Salicylic Acids Enhance Biomass, Total Phenolics Content, and Antioxidant Activity of Adventitious Root Suspension Cultures of Acmella radicans (Jacq.) R.K. Jansen”
Thank you for the opportunity to review this paper regarding the procedure for obtaining in vitro cultures (plants, adventitious roots) of Acmella radicans.
Data from the scientific literature indicate that folk applications may be conditioned by the content of secondary metabolites mentioned by the authors of the work.
Point 1. The legitimacy of undertaking research in in vitro cultures makes sense in the case of relict plants, protected rare plants, etc. Does the species studied by the authors of the manuscript belong to this category of plants? This was not mentioned in the justification for undertaking the research. However, regardless of whether this plant is rare or more common in the world, the developed in vitro cultures are of practical importance because - among other things, it gives procedures for introducing other, endangered or rare plant species from this genus or closely related to in vitro cultures, and it is a form of safeguarding genetic resources.
Dear Editors, Authors. Before the work goes to print, please respond to my comments point by point. Please pay particular attention to errors in terminology regarding the concept of suspension cultures.
Other comments are included in the form of a commentary in the text (pdf).
Response 1. According to the Mexican Official Standard "NOM-059-SEMARNAT-2010, Environmental protection-Mexican native species of wild flora and fauna-Risk categories and specifications for their inclusion, exclusion or change-List of species at risk", Acmella radicans is not listed in any risk category. However, using biotechnological applications is a viable alternative for the sustainable use of medicinal plants, thus conserving our genetic resources. This was added in the manuscript. Lines 51-54. See Link: https://www.profepa.gob.mx/innovaportal/file/435/1/NOM_059_SEMARNAT_2010.pdf
Point 2. Dear authors. I would like to ask you to change the title and the precision and correct use of terms from plant biotechnology. Please check what is meant by the term "suspension culture". You can check this in any biotechnology textbook. A cell suspension or suspension culture is -a type of cell culture in which single cells or small aggregates of cells are allowed to function and multiply in an agitated growth, thus forming a suspension.
Response 2. Yes, that is correct when the term is applied to microorganisms or plant cell cultures growing in a liquid culture medium.
Cell suspension culture: a plant cell suspension culture is a sterile (closed) system normally, initiated by aseptically placing friable callus fragments into a suitable sterle liquid medium.
Suspension culture: type of cell culture in which single cells or small aggregates of cells are allowed to function and mutiply in an agitated growth medium, thus forming a suspension.
Ref. Kumar, G. S.; Kuo, C.-L.; Chang, H.-L.; Chan, H.-S.; Chen, E.C.-F.; Chueh, F.-F.; Tsay, H.-S. In vitro propagation and approaches for metabolites production in medicinal plants. 2012. Advanves in Botanical Research, Vol. 12, 36-55.
However, the term “suspension culture” is accepted in plant biotechnology when culturing organs (anthers, shoots, or roots), since they remain suspended or into the liquid medium [George, E. F., Hall, M. A., & De Klerk, G. J. (Eds.). (2007). Plant propagation by tissue culture: volume 1. the background (Vol. 1). Springer Science & Business Media] (Pages 1-4, Chap. 1 and page 425, Chapter 12). Several works have also been reported using the term “roots suspension cultures”, for example: https://doi.org/10.1016/j.plantsci.2006.03.005; https://doi.org/10.1016/j.indcrop.2019.05.007; https://doi.org/10.1007/s11738-011-0837-2. But the “Shake Flasks” method is also used, for example: DOI 10.1007/s10529-014-1695-y; https://doi.org/10.1016/S0922-338X(97)89266-5; https://doi.org/10.3390/molecules22060880; https://doi.org/10.1016/j.procbio.2005.03.071.
Therefore, we decided to modify the title, which is currently called “Jasmonic and Salicylic Acids Enhance Biomass, Total Phenolics Content, and Antioxidant Activity of Adventitious Roots of Acmella radicans (Jacq.) R.K. Jansen Cultured in Shake Flasks.”.
The entire manuscript was also reviewed and modified where necessary.
Point 3. The specimen 209659 code could not be found in the indicated herbarium.
Response 3. Thanks for the observation. We have contacted the Herbarium staff and the database has been updated. The specimen No. 209659 can now be seen in the following link: https://swbiodiversity.org/seinet/collections/individual/index.php?occid=31021853
Point 4. Please insert the full name in parentheses after the first use of the abbreviation.
Response 4. “Plant Growth Regulators (PGRs)” is previously mentioned in the Introduction section. Lines 61, 62.
Point 5. solid MS medium
Response 5. “gelled MS medium” was substituted by “solid MS medium”. Line 99.
Point 6. Mistake? 0.01 mg?
Response 6. There was a mistake. “0.0” was deleted. The phrase “A control without PGRs was also used” was added. Lines 100, 101.
Point 7. It wasn't a suspension culture. See note at the beginning of the manuscript!
Response 7. This has been corrected. “Suspension cultures” were changed to “shake flasks” or “shake flasks cultures”, were necessary. Line 105, 106.
Point 8. It wasn't a suspension culture. Change according to my comment at the beginning of the manuscript!
Response 8. This has been corrected. “Suspension cultures” were changed to “shake flasks” or “roots culture in shake flasks”, were necessary. Line 108.
Point 9. Please explain what the concentrations in brackets contain and how they relate to the values. “330 μL extracts (1.56 – 100 µg/mL)”.
Response 9. For each concentration of extract or ascorbic acid, respective stock solutions were prepared to obtain the final concentration required. The sentence was rewritten. Line 181 and 183.
Point 10. “[30]. In fact, it has been reported that high auxin/cytokinin ratio or only auxin favors the presence of callus [31]. Other reports indicate that there may also be the simultaneous formation of roots and callus and it depend on the type of explant and the growth regulator [32‒34].” This is common knowledge, not relevant to the research question of this manuscript, and may be omitted.
Response 10. We reviewed and decided not to omit the paragraph as other reviewers commented on it.
Point 11. Figure caption 2. Gelled MS medium
Response 11. “gelled” was substituted by “solid”. Line 228.
Point 12. Figure caption 3. See general note on "suspension culture".
Response 12. The Figure caption 3 was modified. Line 305.
Point 13. Figure caption 4. See general note on "suspension culture".
Response 13. The Figure caption 4 was modified. Line 339.
Point 14. Figure caption 5. See general note on "suspension culture".
Response 14. The Figure caption 5 was modified. Line 373.
Point 15. Figure caption 6. See general note on "suspension culture".
Response 15. The Figure caption 6 was modified. Line 396

Round 2
Reviewer 1 Report
Authors have addressed all my queries and incorporated the same in the revised manuscript. Now the manuscript sounds good. Therefore I recommend the manuscript can be accepted for publication in Biomolecules